

# Effects of plastic mulching on the accumulation and distribution of macro and micro plastics in soils of two farming systems in Northwest China

Fanrong Meng[1,2], Tinglu Fan[3], Xiaomei Yang[1,4], Michel Riksen[1], Minggang Xu[2] and Violette Geissen[1]

[1] Soil Physics and Land Management, Wageningen University, Wageningen, The Netherlands
[2] National Engineering Laboratory for Improving Quality of Arable Land, Institute of Agricultural Resources and Regional Planning, Chinese Academy of Agricultural Sciences, Beijing, China
[3] Dryland Agriculture Institute, Gansu Academy of Agricultural Sciences, Lanzhou, Gansu, China
[4] College of Natural Resources and Environment, Northwest A&F University, Yangling, Shaanxi, China

Corresponding authors
Xiaomei Yang, xiaomei.yang@wur.nl
Minggang Xu, xuminggang@caas.cn

## ABSTRACT

**Background:** Inappropriate disposal of the plastic mulching debris could create macroplastics (MaPs) and microplastics (MiPs) pollution in agricultural soil.
**Methods:** To study the effects of farming systems on accumulation and distribution of agricultural plastic debris, research was carried out on two farming systems in Northwest China. Farming in Wutong Village (S1) is characterized by small plots and low-intensity machine tillage while farming in Shihezi (S2) is characterized by large plots and high-intensity machine tillage. In September 2017, we selected six fields in S1, three fields with 6–8 years of continuous plastic mulching (CM) as well as three fields with over 30 years of intermittent mulching (IM). In S2, we selected five cotton fields with 6, 7, 8, 15 and 18 years of continuous mulching. In both regions, MaPs and MiPs from soil surface to 30 cm depth (0–30 cm) were sampled.
**Results:** The results showed that in S1, MaPs mass in fields with 6–8 years CM (i.e., 97.4kg·ha$^{-1}$) were significantly higher than in fields with 30 years IM (i.e., 53.7 kg·ha$^{-1}$). MaPs in size category of 10–50 cm$^2$ accounted for 46.9% in fields of CM and 44.5% in fields of IM of total collected MaPs number. In S2, MaPs mass ranged from 43.5 kg·ha$^{-1}$ to 148 kg·ha$^{-1}$. MaPs in size category of 2–10 cm$^2$ account for 41.1% of total collected MaPs number while 0.25–2 cm$^2$ accounted for 40.6%. MiPs in S1 were mainly detected in fields with over 30 years of intermittent mulching (up to 2,200 particles·kg$^{-1}$ soil), whereas in S2 were detected in all fields (up to 900 particles·kg$^{-1}$ soil). The results indicated farming systems could substantially affect the accumulation and distribution of agricultural plastic debris. Continuous plastic mulching could accumulate higher amount of MaPs than intermittent plastic mulching. High-intensity machine tillage could lead to higher fragmentation of MaPs and more severe MiPs pollution. These results suggest that agricultural plastic regulations are needed.

## INTRODUCTION

Plastic mulching is a widespread agricultural practice in arid and semi-arid agricultural areas. Plastic mulching has been proved to be beneficial in conserving water (*Ingman, Santelmann & Tilt, 2017*), increasing surface soil temperature, modifying microclimates (*Tarara, 2000*), reducing weeds, discouraging pests (*Díaz-Hernández & Salmerón, 2012*), and improving crop productivity (*Scarascia-Mugnozza, Sica & Russo, 2011*). Plastic mulching has experienced a rapid growth in China since it was first introduced in the 1980s (*Cai et al., 2013*; *Ma et al., 2018*), growing from 6,000 tons used on 117,000 hectares of land in 1982 to about 1.5 million tons used on 18.4 million hectares of land in 2016 (*NBSC, 2017*). Due to the high labour-intensity and costs of removal, plastic films were usually left in fields after crops were harvest. There is a growing concern about the impacts of these discarded plastics on soil health and food security (*Blanco et al., 2018*; *Briassoulis et al., 2013*).

Macro-size plastics (MaPs) in agricultural fields have been reported could significantly reduced the gravimetric water mass and bulk density of soils, decreases macro-pores and alters soil water distribution (*Jiang et al., 2017*). *Zhang et al. (2017)* indicated that soil enzyme activity and soil fertility could be significantly decreased when plastic debris mass reached up to 450 kg·ha$^{-1}$. Plastic debris may act as potential pesticide vehicles in soil and lead to unpredictable migration of pesticides in the soil matrix (*Ramos et al., 2015*; *Teuten et al., 2009*). Furthermore, agricultural plastic mulching has been reported as a source of microplastics (MiPs) in terrestrial environment (*De Souza Machado et al., 2018*; *Huang et al., 2020*; *Scheurer & Bigalke, 2018*). *Rillig (2012)* reported that MiPs could be ingested by soil mesofauna and microfauna and thus, bio-accumulate in the food chain. MiPs could also negatively affect the growth and survival rate of soil organisms and influence soil function (*Cao et al., 2017*; *Huerta Lwanga et al., 2016*). In addition, plastic debris could be easily migrated into surrounding ecosystems (*Rezaei et al., 2019*; *Vermeiren, Munoz & Ikejima, 2016*). It has been widely reported that plastic debris poses considerable threats by choking and starving wildlife (*Barnes et al., 2009*) and by transferring and releasing chemicals into aquatic ecosystems (*Teuten et al., 2009*). Hence, it is of vital importance to monitor the dynamic of plastic debris.

Previous research that documented agricultural plastic debris accumulation mainly attributed it to the mulching time. *Ma & Yang (2013)* reported that plastic debris accumulated in Xinjiang fields at a rate of 27.6, 30.8 and 42.3 kg·ha$^{-1}$ with <10, 10–20 and 20–30 years of mulching, respectively. *He et al. (2018)* observed that the annual rate of plastic debris accumulation was 15.69 kg·ha$^{-1}$ in Xinjiang. However, other factors such as the size of plastic debris, continuous or intermittent mulching and debris recycling activities could also affect the accumulation of MaPs in agricultural soils (*Briassoulis et al., 2004*; *Ma & Yang, 2015*; *Qi et al., 2020*; *Steinmetz et al., 2016*). In different farming regions, different farming practices (mechanical tillage intensity, plastic mulching techniques, etc.) were applied due to the local soil type and climate, thus resulting in different accumulation patterns of plastic debris. *Yan et al. (2008)* conducted a field observation in Xinjiang (Northwest China) and found that highest amount of MaPs reached up to

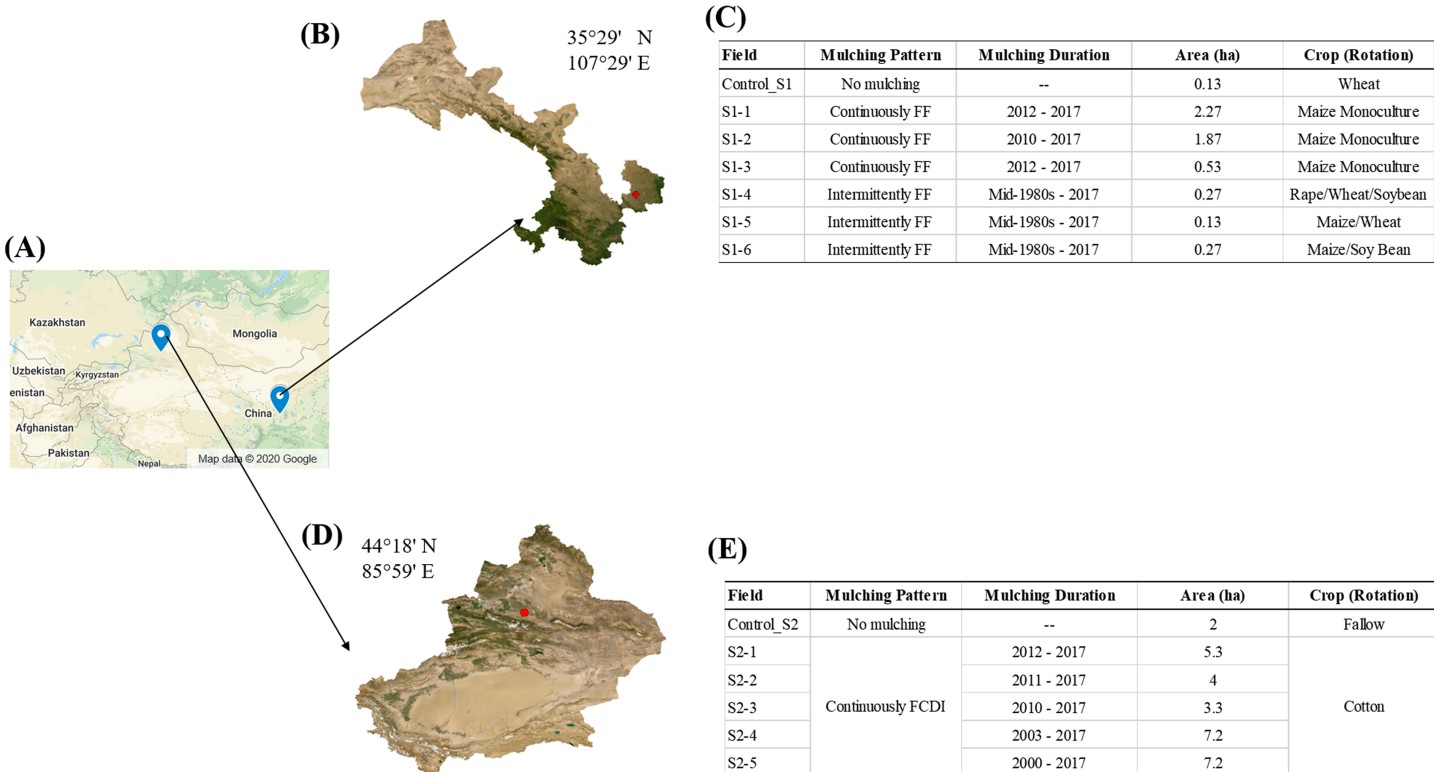

**Figure 1 Sampling site.** (A) Field measurements were conducted in two regions in Northwest China (Map data © 2020 Google). (B) Study site Gansu (S1), was characterized by small plots and low intensity tillage. (C) Mulching pattern, duration, field area and crop rotation of the selected fields in S1. (D) Study site Xinjiang (S2), was characterized by large plots, lower plastic input and intensive machine tillage. (E) Mulching pattern, duration, field area and crop rotation of the selected fields in S2.

308 kg·ha$^{-1}$, MaPs were mainly concentrated in 0–10 cm soil. They also found that 80% of MaPs detected in their study were in the size category of 1–25 cm$^2$. *Li et al. (2017)* conducted a field observation in Qingdao (Middle China) and found the amount of MaPs in agricultural fields was ranging between 11-69 kg·ha$^{-1}$. MaPs were mainly concentrated in 0–20 cm soil. However, in their study, the detected MaPs were mainly in the size category of >100 cm$^2$. Therefore, farming system plays an important role in agricultural plastic pollution. Unfortunately, the effects of different farming systems on plastic accumulation remained inadequate addressed.

In this current work, we assumed that different farming systems could affect the accumulation and distribution of plastic debris in agricultural soil. We hypothesized that (1) Under the same farming system, continuous plastic mulching could accumulate more MaPs mass than intermittent plastic mulching; (2) farming system of higher mechanical intensity could lead to higher fragmentation of MaPs and create more MiPs than farming system of lower mechanical intensity. To test our hypothesis, we selected two regions in Northwest China that both have a long history (dating back to the mid-1980s) of plastic mulching application but with different farming systems (Fig. 1). First study region is characterized by small-scale farmlands with low levels of agricultural mechanization. Second study region is characterized by large-scale farmlands and high levels of

agricultural mechanization. We examined the accumulation and distribution of MaPs and MiPs in 0–30 cm soil of two study regions. In our paper, MaPs were defined as plastic particles with a size area of $>0.25$ cm$^2$ (which was the smallest MaPs size we collected from field, Fig. S3A). MiPs were defined as plastic particles derived from LDPE plastic mulching film with a diameter of $<2$ mm and a density smaller than 1 g·cm$^{-3}$ due to plastic mulching was considered as the main source for plastic pollution in the selected two study regions. We hope to provide a basis information for future efforts aimed at controlling and managing plastic pollution in agricultural soils.

# MATERIALS AND METHODS

## Study area description

The first study region located in Wutong Village (S1, 35°29′ N, 107°29′ E), Gansu Province, where the cultivated area is 2,779 ha. S1 is characterized by small-scale farmlands (usually smaller than 1 ha according to farmers) and low levels of agricultural mechanization. Tillage is performed using small rotary cultivators at a depth of 30 cm and harvesting is mainly done manually. In S1, farmers predominantly practiced full film flat mulching (FF, Fig. 2A). The plastic film was transparent and made from low-density polyethylene (LDPE) (Table S1) and there was an annual usage of 150 kg·ha$^{-1}$ in this area. Plastic mulching had been intermittently applied to the fields over a span of 30 years at 3- or 4-year intervals. Maize *(Zea mays L.)* was the main crop for which plastic mulching was used. After the maize had been sown, the land was covered with the plastic much. The maize plants grew through the plastic mulch. After harvesting the maize, the plastic films were manually removed from the soil before preparing the land for the next crop. The common practice in study area S1 was to rotate maize with soybean *(Glycine max)*, oilseed rape *(Brassica napus)* and winter wheat *(Triticum aestivum L.)*. For these other crops, plastic mulching was not used. The common cultivation pattern was three harvests every two years. However, in recent years, some farmers have switched to a monoculture of maize due to its increasing economic value. In S1, we selected 6 fields to investigate the impacts of monoculture and crop rotation on agricultural plastic debris accumulation and distribution (Fig. 1A). Fields S1-1 (contact: Shangzhong Li), S1-2 (contact: Yi Dang) and S1-3 (contact: Lei Wang) were monocultured with maize, with 6, 8 and 6 years of continuous mulching, respectively. In fields S1-4, S1-5 and S1-6, crops were rotated. In S1-4 (contact: Limin Wang), the crop rotation was oilseed rape (Early September 2015 to mid-June 2016), Winter Wheat (late September/early October 2017 to the end of May 2018), and Soybean (mid-June 2017 to late September 2017). In S1-5 (contact: Sanzhi LI), the crop rotation was Maize (mid-April 2016 to mid-September 2016) and Winter Wheat (late September/early October 2017 to the end of May 2018). In S1-6 (contact: Limin Wang), the crop rotation was Maize (mid-April 2016 to mid-September 2016), left Fallow (mid-September 2016 to mid-June 2017), and Soybean (mid-June 2017 to late September 2017).

The second study region located in Shihezi City (S2, 44°18′ N, 85°59′ E), Xinjiang, where the cultivated area is 971,301 ha. S2 is characterized by large-scale farmlands (larger than 3 ha per field) and high levels of agricultural mechanization. Tillage is performed

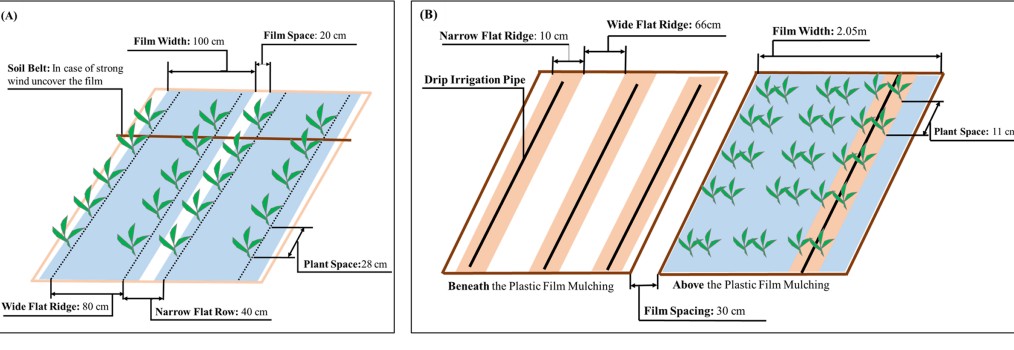

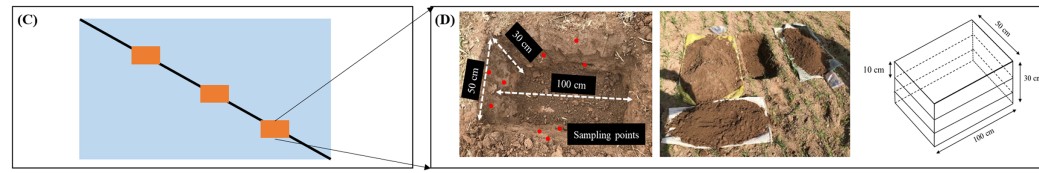

**Figure 2 Schematic of plastic mulching patterns.** (A) Full film flat mulching (FF) in S1. (B) Flat cover with drip irrigation under plastic film (FCDI) in S2. (C) Sampling quadrats location in the field. (D) Sampling activity, the red dots represent the sampling points for extraction of microplastics.

using large rotary cultivators at a depth of 30 cm and harvesting is done using a cotton picker. In S2, the mulching pattern was flat cover combined with drip irrigation (FCDI, Fig. 2B). The use of plastic mulching coupled with irrigation was initiated in the 1990s. The annual usage of plastic film in S2 was 60 kg·ha$^{-1}$. The plastic film was transparent and made from LDPE and LLDPE (linear low-density polyethylene) (Table S1).

We selected five fields with 6 (contact: Yu Liu), 7 (contact: Yu Liu), 8 (contact: Gongmao Wang), 15 (contact: Jihong Shi) and 18 (contact: Jiancheng Liu) years of continuous FCDI mulching (Fig. 1B). All the fields were planted with the same type of cotton cultivar and the same fertilization practices were followed. After harvesting, the plastic films were machinery removed from the fields along with the cotton stalks.

The farming system chosen in each region was representative of the typical situation for the local farmers. Both regions have a temperate continental climate. The climate data from 2017 of two study region and the soil information are shown in Table S1. The climate data was recorded by a local weather station. For this research, we assumed that the mixing procedure mainly depended on the different farming systems prevalent in the two study regions. Hence, only farming systems and crops from these regions were taken into account in current work.

## Field sampling of macroplastics and soil samples

MaPs were manually collected using a quadrat sampling method. In each selected field, on the diagonal line (Fig. 2C), we randomly dug three quadrats (each quadrat was 100 cm long, 50 cm wide and 30 cm deep and covered two crop rows, Fig. 2D). Each sampling quadrat was then separated into three depth layers: 0–10 cm, 10–20 cm and 20–30 cm. The entire soil mass from each layer were then put onto a "flat polypropylene (PP) wire weaved mesh sheet" (Fig. S1). To prevent further fragmentation of the MaPs

during the collection process, visible MaPs were gently picked out by hand from the entire mass of each layer of the sampled quadrat. Each layer of the sampling quadrat was carefully checked three times. The collected MaPs from each layer were then stored in a PP bag. All the collected MaPs samples then transferred to the laboratory for further analysis. Totally, resulted in 54 MaPs samples in S1 (six fields × three quadrats × three layers) and 45 MaPs samples in S2 (five fields × three quadrats × three layers).

## Macroplastic quantification and residual ratio

In the laboratory, all MaPs were cleaned thoroughly. First, plant roots, soil, sand, etc. were separated manually from the MaPs. Next, MaPs were unfolded and washed with tap water three times in a PP basin (solid colour) until the films were transparent. Then, the cleaned MaPs from each layer of soil quadrat were stored in a 500 mL glass beaker that was filled with 300 mL of tap water. The beaker was put into a ultrasonic cleaner (KQ 3200DA, Kunshan Ultrasonic Instruments Co., Ltd, China) for 1 h to remove any fine sands attached to the MaPs. The MaPs then were stored in a PP mesh bag (12 × 15 cm, diameter 1 mm) and air-dried for 2-days. The collected MaPs were present in arbitrary shapes, that is, curved together and flake shape (Fig. S2). All the collected MaPs were gently spread and measured by using graph paper. The smallest size of collected MaPs was measured 0.25 cm² (Fig. S3A). We separate the plastic debris into five size groups: 0.25–2 cm², 2–10 cm², 10–50 cm², 50–100 cm² and >100 cm² (Figs. S3B–S3F). For each size group, MaPs were weighed using an analytical balance (METTLER AE 200, METTLER AE 200, MARSHALL SCIENCE, accuracy of 0.1 mg) and the number of particles (p) was counted. The concentrations of MaPs were recorded as mass (kg·ha$^{-1}$) and number (p·ha$^{-1}$). The mass and number were calculated as follows:

$$(M_i/S) \times 100 = \text{Mass} \, (\text{kg·ha}^{-1}) \tag{1}$$

$$(N_i/S) \times 100 = \text{Number} \, (\text{p·ha}^{-1}) \tag{2}$$

Where, $M_i$ (mg) is the total weight of collected MaPs from each of the 10 cm sampling depths. S (cm²) is the surface area of each sampling quadrat. The conversion coefficient from mg·cm$^{-2}$ to kg·ha$^{-1}$ is 100. $N_i$ (p) is the total number of collected MaPs from each of the 10 cm sampling depths.

The residual ratio of MaPs per selected field, which referred to as the amount of MaPs found in soil in relation to the total applied biofilm amount, was calculated with the following equation:

$$U \times t = \text{Total input} \tag{3}$$

$$\text{Mass/total input} \times 100 = \text{Residual ratio} \, (\%) \tag{4}$$

where, $U$ is the annual plastic film usage, S1 is 150 kg·ha$^{-1}$ and S2 is 60 kg·ha$^{-1}$. $t$ is the number of years of plastic mulching application. Mass is the total weight of MaPs collected from the selected fields. For fields where mulching was used continuously, $t$ is equal to the mulching year. However, for the intermittent use of plastic mulching in the S1 region,

*t* is assumed to be 7 years, which was calculated using the plastic mulching interval of every 4 years during the 30 years of mulching history.

## Microplastic extraction and identification

Soil samples for MiPs extraction were collected from the sides of the pits that were dug out of the soil quadrats (Fig. 2C). For each 10 cm depth, 1 kg of soil sample from three randomly selected sampling points were directly collected in situ and homogenized. The soil samples were extracted using a metal augur and transferred to the laboratory in PP plastic bags (Fig. S4). Once in the lab, the soil samples were air dried in open paper trays in a room without visible plastic materials. The soil was then sieved through a 2 mm steel sieve for the further MiPs extraction. A control field had never been applied with plastic mulching was selected in each study region (Contact of control field in S1: Jianjun Zhang; in S2: Yu Liu). A soil sample from each control field was collected to check whether the PP plastic containers polluted the soil samples with plastic.

MiPs extraction was carried out following a float method published by *Zhang et al. (2018)*. This method was specially developed for the extraction of LDPE-MiPs. According to *Zhang et al. (2018)*, the recovery rates were >90% and the lower limit of detection for this method is 20 μm. Before the extraction procedure, the MaPs in the soil samples were collected until no more plastic could be seen with the naked eye. Then, 10 g of the air-dried soil samples were added to 100 ml centrifuge tubes (PP). A total of 50 mL of distilled water was added to each tube and a glass stick was used to stir the soil and water together in order to get a homogeneous suspension. The glass stirrer was rinsed off using distilled water and the water was then collected in the same centrifuge tube. Next, soil samples were spun four times using high speed centrifugation (GL-21MC/GL21MC, Cence Xiangyi, China) at 14,400×*g* for 10 min to separate the soil particles from the floating materials. The resulting supernatant was filtered using filter paper (pore diameter <3 μm). After that, 50 mL of distilled water was added again to each centrifuge tube and then placed in an ultrasonic cleaner (KQ 3200DA, Kunshan Ultrasonic Instruments Co., Ltd, China) for 2 h in order to isolate any MiPs that might have still been adsorbed on soil micro aggregates. The samples were then centrifuged for a fifth time. Finally, the filter papers (pore diameter <3 μm) with the extracted MiPs were dried in an oven (Type A 1500-145, Kema Keur) at 60 °C to a constant weight and stored in glass Petri dishes for optical inspection. The soil samples from the control fields were also put through the same procedure for extracting MiPs.

As a quality control measure, each set of soil samples (*n*) from each study site (*n* = 18 in S1, *n* = 15 in S2) contained three blank samples of distilled water. This measurement was used to account for any contamination which could have occurred inside the lab (*Mahon et al., 2017*; *Scheurer & Bigalke, 2018*). White cotton lab coats were worn during analysis and sample manipulations.

The extracted MiPs were inspected using a microscope (Leica wild M3C, Type S, simple light) at 6.4 X Zoom. The MiPs collected from each filter were placed on glass slides. The glass slides were then inspected using a microscope and a picture "*I*" was taken. In order to get rid of any organic material from the soil samples that might have interfered
with the counting, the glass slides were placed gently on top of an electric heating plate. (Type A 1500-145, Kema Keur) and heated for 5–7 s at 130 °C in order to melt the MiPs. The MiPs were transformed into transparent shiny surfaces which could be easily distinguished from soil particles. The glass slides were then inspected again using the microscope and a second picture "*II*" was taken. By comparing pictures "*I*" and "*II*", the melted MiPs could be identified. The smallest microplastic particle detected in our work was 0.49 mm (44 pixels), calculated by image J, 1 pixel = 0.585/60 mm (microscope at 6.4 X Zoom). The picture of the setup for identification MiPs is presented in Fig. S5.

### Limitation of microplastic extraction method

Only MiPs from LDPE or MiPs with density $<1$ g·cm$^{-3}$ were able to be extracted due to the water reagent. MiPs with densities higher than 1 g·cm$^{-3}$ (e.g., PVC 1.45 g·cm$^{-3}$) were not able to be extracted (*Nuelle et al., 2014*). However, this method provides a validated method for estimating the presence of LDPE-MiPs in the soil.

### Data analysis

The arcsine square root transformation was applied to the mass and number of MaPs pieces to avoid violating the underlying assumptions of normality. One-way analyses of variances (ANOVAs) were applied to compare the mass and number of MaPs pieces between different fields within the same selected region and different soil layers within the same field, followed by the application of an LSD post hoc test at the $p < 0.05$ level. MaPs mass (kg·ha$^{-1}$) and number (p·ha$^{-1}$) were presented as "means ± standard deviations". MiPs that were detected in the fields were presented in raw data in the unit of p·kg$^{-1}$ soil due to the highly random distribution of the particles and no statistical test was performed.

## RESULTS

### Accumulation and distribution of macroplastics in selected agricultural fields

In S1, across the 6 selected fields, MaPs number varied from $56.7 \times 10^4$ p·ha$^{-1}$ to $264.7 \times 10^4$ p·ha$^{-1}$ and MaPs mass varied from 53.7 kg·ha$^{-1}$ to 108 kg·ha$^{-1}$ (Table 1). Fields with 6–8 years of continuous plastic mulching use (S1-1, S1-2 and S1-3) showed significant higher MaPs number (one-way ANOVA, $F_{5, 12} = 20.9$, $p < 0.01$) and MaPs mass (one-way ANOVA, $F_{5, 12} = 4.24$, $p = 0.02$) than fields with more than 30 years of intermittent plastic mulching use (S1-4, S1-5), except S1-6, where fields showed similar numbers of MaPs as compared to S1-2. The residual ratios varied from 5.11% to 12.0% across the selected fields (Table 2). Fields where continuous mulching were practiced (S1-1, S1-2 and S1-3) showed significantly higher residual ratios as compared to fields with intermittent mulching (S1-4, S1-5 and S1-6) (Table 2, one-way ANOVA, $F_{5, 12} = 6.89$, $p = 0.03$). The distribution patterns of MaPs in each 10 cm of 0–30 cm soil layer across the 6 fields in S1 are presented in Table 3. The results showed that MaPs were mainly concentrated in the 0–10 cm soil layer, followed by 10–20 cm, and then 20–30 cm. The number of MaPs in the 0–10 cm layer was significantly higher (one-way ANOVA, $p < 0.01$, Table S6)

**Table 1 Macroplastics number and content in 0–30 cm (one way ANOVA and followed by LSD test at the $p < 0.05$ level).**

| Study region | Sampling Site | MaPs number ($\times 10^4$ p·ha$^{-1}$) | MaPs mass (kg·ha$^{-1}$) |
|---|---|---|---|
| S1 | S1-1 | 235 ± 45.8a | 105 ± 20.1a |
| | S1-2 | 170 ± 40.8b | 97.4 ± 22.0a |
| | S1-3 | 265 ± 12.9a | 108 ± 13.2a |
| | S1-4 | 88.0 ± 22.3c | 56.1 ± 37.3b |
| | S1-5 | 56.7 ± 4.2c | 57.1 ± 16.4b |
| | S1-6 | 155 ± 58.2b | 53.7 ± 12.4b |
| S2 | S2-1 | 502 ± 201c | 43.5 ± 9.3c |
| | S2-2 | 650 ± 136c | 88.9 ± 12.2b |
| | S2-3 | 461 ± 79.1c | 80.6 ± 18.6b |
| | S2-4 | 2,016 ± 188a | 148 ± 28.1a |
| | S2-5 | 991 ± 163b | 81.1 ± 3.93b |

Note:
Lowercase letters (a,b,c) indicate significant difference between different selected fields. In S1: S1-1: 6 years of FF mulching; S1-2: 8 years of FF mulching; S1-3: 6 years of FF mulching; S1-4, S1-5 and S1-6: 30 years history of intermittent FF mulching. In S2: S2-1: 6 years of FCDI mulching; S2-2: 7 years of FCDI mulching; S2-3: 8 years of FCDI mulching; S2-4: 15 years of FCDI mulching; S2-5: 18 years of FCDI mulching.

**Table 2 Residual ratios of MaPs in Wutong Village, Gansu Province (S1) and Shihezi City, Xinjiang Province (S2).**

| Study site | Sampling site | Mulching pattern | Input per application (kg·ha$^{-1}$) | Mulching duration | Total input (kg·ha$^{-1}$) | Collected MaPs (kg·ha$^{-1}$, average) | Residual ratio (%) |
|---|---|---|---|---|---|---|---|
| S1 | S1-1 | Continuously FF | 150 | 2012–2017 | 900 | 105 | 11.7 ± 2.23a |
| | S1-2 | Continuously FF | | 2010–2017 | 1,200 | 97.4 | 8.12 ± 1.83a |
| | S1-3 | Continuously FF | | 2012–2017 | 900 | 108 | 12.0 ± 1.47a |
| | S1-4 | Intermittently FF | | Mid-1980s–2017 | 1,050 | 56.1 | 5.34 ± 3.56b |
| | S1-5 | Intermittently FF | | Mid-1980s–2017 | 1,050 | 57.1 | 5.44 ± 1.56b |
| | S1-6 | Intermittently FF | | Mid-1980s–2017 | 1,050 | 53.7 | 5.11 ± 1.19b |
| S2 | S2-1 | Continuously FCDI | 60 | 2012–2017 | 360 | 43.5 | 12.1 ± 6.76ab |
| | S2-2 | | | 2011–2017 | 420 | 88.9 | 21.2 ± 7.64a |
| | S2-3 | | | 2010–2017 | 480 | 80.6 | 16.8 ± 3.87ab |
| | S2-4 | | | 2003–2017 | 900 | 148 | 16.5 ± 5.62ab |
| | S2-5 | | | 2000–2017 | 1,080 | 81.1 | 7.51 ± 0.36b |

Note:
Total input was calculated using equation 3 and residual ratio was calculated using equation 4. Lowercase letters (a, b, c and d) indicate significant differences between different selected fields within each selected study region ($p < 0.05$, $n = 3$). Removal rate estimated based on farmers' information.

than the 10–20 cm and/or 20–30 cm soil layers. However, for the mass of MaPs, there were significant differences (one-way ANOVA, more detail showed in Table S6) found between soil layers 0–10 cm and 10–20 cm and between soil layers 10–20 cm and 20–30 cm, except for S1-1 and S1-6. In addition, we also compared the number and mass percentage of MaPs in different size categories (Fig. 3; Table S4). Continuous (S1-1, S1-2 and S1-3) and intermittent (S1-4, S1-5 and S1-6) mulching fields showed similar composition patterns. For MaPs number, size category of 10–50 cm$^2$ accounted for highest

**Table 3 Macroplastics number and mass in different soil layer in S1 (one way ANOVA and followed by LSD test at the $p < 0.05$ level).**

| Sampling site | Soil layer | MaPs number ×10⁴ (p/ha) | MaPs mass kg·ha⁻¹ |
|---|---|---|---|
| S1-1 | 0–10 cm | 134 ± 25.1a | 70.1 ± 1.98a |
| | 10–20 cm | 78.7 ± 37.2a | 30.5 ± 15.9b |
| | 20–30 cm | 22.7 ± 15.5b | 4.26 ± 3.69c |
| S1-2 | 0–10 cm | 111 ± 38.2a | 64.7 ± 12.7a |
| | 10–20 cm | 43.3 ± 13.3b | 28.2 ± 16.5b |
| | 20–30 cm | 16.0 ± 12.2b | 4.50 ± 3.19c |
| S1-3 | 0–10 cm | 173 ± 27.3a | 82.3 ± 10.6a |
| | 10–20 cm | 69.3 ± 21.9b | 17.9 ± 2.25b |
| | 20–30 cm | 22.0 ± 11.1c | 8.13 ± 7.93b |
| S1-4 | 0–10 cm | 50.0 ± 10.4a | 42.2 ± 19.5a |
| | 10–20 cm | 33.3 ± 29.1a | 13.0 ± 17.2b |
| | 20–30 cm | 4.67 ± 3.06b | 0.82 ± 0.77b |
| S1-5 | 0–10 cm | 30.0 ± 7.2a | 33.6 ± 18.4a |
| | 10–20 cm | 19.3 ± 2.31a | 14.7 ± 1.11ab |
| | 20–30 cm | 7.33 ± 5.03b | 6.79 ± 6.97b |
| S1-6 | 0–10 cm | 71.3 ± 17.9a | 30.2 ± 5.63a |
| | 10–20 cm | 53.3 ± 5.03ab | 16.1 ± 1.42b |
| | 20–30 cm | 30.7 ± 19.0b | 7.30 ± 5.84c |

**Note:**
Lowercase letters (a, b and c) indicate significant difference between different layers within same selected field. S1-1: 6 years of FF mulching; S1-2: 8 years of FF mulching; S1-3: 6 years of FF mulching; S1-4, S1-5 and S1-6: 30 years history of intermittent FF mulching.

of the total collected MaPs number (46.9% for continuous mulching fields and 44.5% for intermittent mulching fields). Size category of 0.25–2 cm² accounted for lowest of the total collected MaPs number (3.55% for continuous mulching fields and 4.20% for intermittent mulching fields) (Figs. 3A and 3B). Significant differences were observed between different size groups (one-way ANOVA, $F_{4, 40} = 148$, $p < 0.01$ for continuous mulching fields; $F_{4, 40} = 35.9$, $p < 0.01$ for intermittent mulching fields). As for MaPs mass, MaPs in size categories >100 cm² and 10–50 cm² contributed highest (34.8% and 35.8% in continuous mulching fields; 42.9% and 34.2% in intermittent mulching fields) to the total mass while size category of 0.25–2 cm² contributed lowest (0.16% for continuous mulching fields and 0.26% for intermittent mulching fields) (Figs. 3D and 3E). Significant differences were observed between different MaPs size groups (one-way ANOVA, $F_{4, 40} = 217$, $p < 0.01$ for continuous mulching fields; $F_{4, 40} = 28.4$, $p < 0.01$ for intermittent mulching fields).

In S2, across the selected fields, MaPs number varied from 461 × 10⁴ p·ha⁻¹ to 2,016 × 10⁴ p·ha⁻¹ and MaPs mass varied from 43.5 kg·ha⁻¹ to 148 kg·ha⁻¹ (Table 1). Fields exposed to 15 years of plastic mulching use (S2-4) showed significant higher MaPs number (one-way ANOVA, $F_{4, 10} = 61.7$, $p < 0.01$) and mass (one-way ANOVA, $F_{4, 10} = 17.1$, $p < 0.01$) than other selected fields. The residual ratios varied from 7.51% to 21.2%

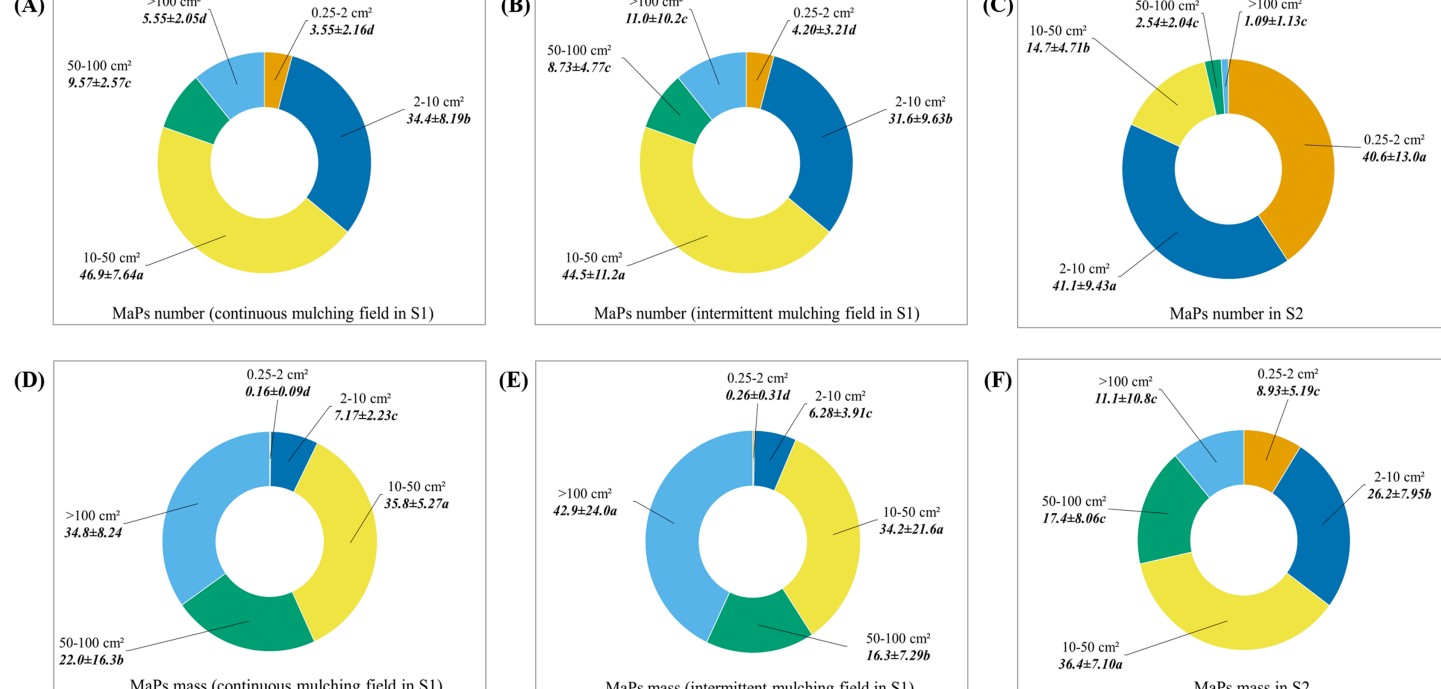

**Figure 3 The percentages of number and mass of MaPs in different size categories.** (A) The number percentages of MaPs in different size categories in 6–8 years continuous mulching fields (S1-1, S1-2 and S1-3) in S1. (B) The number percentages of MaPs in different size categories in over 30 years of intermittent mulching fields (S1-4, S1-5 and S1-6) in S1. (C) The number percentage of MaPs in different size categories in S2. (D) The mass percentage of MaPs in different size categories in continuous mulching fields (S1-1, S1-2 and S1-3) in S1. (E) The mass percentage of MaPs in different size categories in continuous mulching fields (S1-4, S1-5 and S1-6) in S1. (F) The mass percentage of MaPs in different size categories in S2. Lowercase letters (a, b, c and d) indicate significant differences between different size groups ($p < 0.05$).

(Table 2). The field exposed to 18 years of plastic mulching use (S2-5) showed the lowest residual ratio, which was only significantly lower than field S2-2 (Table 2, one-way ANOVA, $F_{4,\,10} = 2.68$, $p = 0.09$). The distribution patterns of MaPs in each 10 cm soil 0–30 cm across the five fields in S2 are presented in Table 4. MaPs were mainly concentrated in the first 0–10 cm soil layer, followed by 10–20 cm and 20–30 cm. For the numbers of MaPs, the significant differences were mainly found between the 0–10 cm and 20–30 cm soil layers (one-way ANOVA, one-way ANOVA, more detail showed in Table S7). For the mass of MaPs, the significant differences (one-way ANOVA, more detail showed in Table S7) were found between soil layers 0–10 cm and 10–20 cm and between soil layers 10–20 cm and 20–30 cm, except for S2-1 and S2-5. In S2, we also compared the number and mass percentage of MaPs in different size categories (Fig. 3; Table S4). For MaPs number (Fig. 3C), the highest contributors were size categories of 0.25–2 cm² (40.6%) and 2–10 cm² (41.1%). The lowest contributor was size category of >100 cm² (1.09%). Significant differences were observed between different groups (one-way ANOVA, $F_{4,\,70} = 18.4$, $p < 0.01$). For MaPs mass (Fig. 3F), the highest contributor was size category of 10–50 cm² (36.4%), the lowest contributor was size category of 0.25–2 cm² (26.2%). The significant differences between each group were observed (one-way ANOVA, $F_{4,\,70} = 172$, $p < 0.01$).
**Table 4 Macroplastics number and content in different soil layer in S2 (one way ANOVA and followed by LSD test at the $p < 0.05$ level).**

| Sampling site | Soil layer | MaPs number $\times10^4$ (p/ha) | MaPs mass $kg\cdot ha^{-1}$ |
|---|---|---|---|
| S2-1 | 0–10 cm | 241 ± 24a | 21.4 ± 1.3a |
| | 10–20 cm | 159 ± 72.5ab | 11.6 ± 4.53b |
| | 20–30 cm | 101 ± 44.6b | 10.4 ± 5.05b |
| S2-2 | 0–10 cm | 278 ± 47a | 51.7 ± 4.6a |
| | 10–20 cm | 216 ± 55.2ab | 25.4 ± 8.03b |
| | 20–30 cm | 156 ± 40.8b | 11.7 ± 5.21c |
| S2-3 | 0–10 cm | 251 ± 68.2a | 52.0 ± 9.07a |
| | 10–20 cm | 162 ± 118ab | 24.7 ± 14.6b |
| | 20–30 cm | 48 ± 29.5b | 3.86 ± 3.03c |
| S2-4 | 0–10 cm | 1,011 ± 185a | 78.1 ± 15.2a |
| | 10–20 cm | 685 ± 41b | 50.9 ± 8.13b |
| | 20–30 cm | 320 ± 76c | 19.1 ± 11.2c |
| S2-5 | 0–10 cm | 467 ± 34.5a | 49.6 ± 5.00a |
| | 10–20 cm | 336 ± 130ab | 20.9 ± 9.91b |
| | 20–30 cm | 188 ± 87.1b | 10.6 ± 4.77b |

**Note:**
Lowercase letters (a,b,c) indicate significant difference between different layers within same selected field. S2-1: 6 years of FCDI mulching; S2-2: 7 years of FCDI mulching; S2-3: 8 years of FCDI mulching; S2-4: 15 years of FCDI mulching; S2-5: 18 years of FCDI mulching.

**Table 5 Microplastics (MiPs) number $p\cdot kg^{-1}$ soil in S1.**

| Soil layer | Replicates | Control | S1-1 | S1-2 | S1-3 | S1-4 | S1-5 | S1-6 |
|---|---|---|---|---|---|---|---|---|
| 0–10 cm | 1 | nd | nd | nd | nd | nd | 200 | 200 |
| | 2 | nd | 200 | nd | nd | nd | 100 | 800 |
| | 3 | nd | nd | nd | nd | 200 | nd | nd |
| 10–20 cm | 1 | nd | nd | nd | nd | nd | nd | 200 |
| | 2 | nd | nd | nd | nd | nd | nd | nd |
| | 3 | nd | nd | nd | nd | 100 | nd | nd |
| 20–30 cm | 1 | nd | 1 | nd | nd | 100 | 2,200 | 1,000 |
| | 2 | nd | nd | nd | nd | 100 | nd | nd |
| | 3 | nd | 100 | nd | nd | nd | 200 | nd |

**Note:**
S1-1: 6 years FF mulching; S1-2: 8 years FF mulching; S1-3: 6 years FF mulching; S1-4, S1-5 and S1-6: 30 years history of intermittent FF mulching.

## Occurrence of microplastics in agricultural soils

In two study regions, the occurrence of MiPs was highly random and only the raw data were presented in the form of $p\cdot kg^{-1}$. No MiPs were detected in the control sites and quality controls.

In S1, MiPs were mainly detected in intermittent mulching fields (Table 5). In continuous mulching fields, MiPs were only detected in S1-1, no MiPs were detected in S1-2 or S1-3. The highest MiPs concentration of 2200 $p\cdot kg^{-1}$ was detected in the 20-30 cm

**Table 6 Microplastics (MiPs) number p·kg$^{-1}$ soil in S2.**

| Soil layer | Replicates | Control | S2-1 | S2-2 | S2-3 | S2-4 | S2-5 |
|---|---|---|---|---|---|---|---|
| 0-10 cm | 1 | nd | 600 | 300 | 300 | nd | 900 |
| | 2 | nd | nd | 400 | nd | 100 | 700 |
| | 3 | nd | nd | nd | 900 | 800 | 400 |
| 10–20 cm | 1 | nd | nd | 100 | nd | nd | 800 |
| | 2 | nd | nd | 800 | nd | 400 | 100 |
| | 3 | nd | nd | nd | 100 | 400 | 100 |
| 20–30 cm | 1 | nd | 100 | nd | nd | nd | 300 |
| | 2 | nd | nd | 600 | nd | nd | 300 |
| | 3 | nd | 100 | nd | 200 | 700 | 200 |

**Note:**
S2-1: 6 years FCDI mulching; S2-2: 7 years FCDI mulching; S2-3: 8 years FCDI mulching; S2-4: 15 years FCDI mulching; S2-5: 18 years FCDI mulching. nd = not detected

layer of S1-5, which with 30 years intermittent mulching history. In S2, MiPs were detected in all the selected fields while not all the soil layers (Table 6). In S2-5, MiPs were detected in all the soil samples. The highest MiPs concentration (900 p·kg$^{-1}$) was detected in the 0–10 cm soil layer of S2-3 (8 years mulching) and S2-5 (18 years mulching).

# DISCUSSION

In current research, we aimed to examine the characteristics of the MaPs and MiPs accumulation and distribution under two farming systems. Many previous research attributed the accumulation solely to the mulching year (*He et al., 2018*; *Ma & Yang, 2015*). Understanding the impacts from other factors of different farming system is essential for regulating agricultural plastic film management. However, relevant knowledge is still limited.

## Accumulation and distribution of macroplastics in agricultural soils

In S1, fields with 6–8 years of continuous mulching (S1-1, S1-2 and S1-3) contained significant higher MaPs numbers and mass than fields with 30 years of intermittent mulching (S1-4, S1-5 and S1-6). One possible explanation might be attributed to the removal activity by farmers. In S1, according to local farmers, 80% of applied plastic films (remained intact and could be easily collected) were manually removed after the harvesting of mulched crop and before sowing of the next rotated crop. In addition, the remained smaller particles (could still be picked up by hand) were constantly collected during the seedling and weeding stages. Therefore, fields with 30 years of intermittent mulching, as compared to fields with 6–8 years of continuous mulching, were subject to more plastic debris removal activities. On the contrary, the plastic films in continuous mulching fields were only collected once after the harvest of maize. The smaller particles were remained in soils and experienced freeze-thaw cycles during the winter and spring, which also posed more difficulties for manually removal. As a result, fields with 30 years intermittent mulching accumulated fewer MaPs than continuous mulching fields.

Another possible explanation for this might be attributed to wind dispersion. *Zylstra (2013)* provided evidence that wind action could spread substantial plastic debris between different ecosystems. Strong winds are very common in the Gansu province (*Guan et al., 2017*). Hence, in the fields with over 30 years of intermittent mulching, wind could have dispersed more agricultural plastic debris into other environments and thus, lead to the significant lower accumulation of MaPs.

Crop rotation could have also affected the accumulation of MaPs. Looking closer at our results of fields with 30 years intermittent mulching history in S1, S1-6 showed a higher number of MaPs than S1-4 and S1-5. According to farmers, in S1-4, no plastic mulching was applied to the field from 2015 to 2017 due to the crop rotation of oilseed rape (Early September 2015 to mid-June 2016), winter Wheat (late September/early October 2016 to the end of May 2017), and soybean (mid-June 2017 to late September 2017). In S1-5, the rotation of maize (mid-April 2017 to mid-September 2017, when plastic film was applied) and winter Wheat (late September/early October 2017 to the end of May 2018) required farmland to be ploughed in September. However, in S1-6, with a rotation of maize (mid-April 2016 to mid-September 2016, when plastic was applied), left fallow (mid-September 2016 to mid-June 2017) and Soybean (mid-June 2017 to late September 2017), farmland was ploughed in May while still some plastic debris incorporated into the soil. In addition, the winter could have accelerated the weathering and aging of MaPs. Any of these factors could have led to the higher MaPs number seen in S1-6 as compared to S1-4 and S1-5.

In S2, we detected lower MaPs mass (ranging from 43.5 kg·ha$^{-1}$ to 148 kg·ha$^{-1}$) as compared to other researches. In the same study region, *Yan et al. (2008)* discovered plastic residues of 259.9 kg·ha$^{-1}$ (10 years) and 307.9 kg·ha$^{-1}$ (20 years) in the soils of monocultural cotton. *He et al. (2018)* found that plastic residues (LDPE, LLDPE) ranged from 121.9 to 352.4 kg·ha$^{-1}$ in fields where there were 5–19 years of mulching use. This discrepancy might be explained by the differences between the plastic debris sampling methods. In our field observations, the plastic films found on the surface of the soils were not taken into consideration for measurements since farmers claimed that these films would normally be removed along with the cotton stalk. In addition, in S2, the collected MaPs number and mass were not linearly increased with the mulching year, the highest accumulation was observed at 15 years mulching field. This emphasized that year of mulching use was not the main factor affecting the accumulation of agricultural plastic debris (*Dong et al., 2015*; *Yan et al., 2014*). As we mentioned in "Material and Method", fields in Xinjiang were subjected to high intensity machinery tillage, which lead to higher fragmentation of MaPs (Fig. 3). The smaller particles were difficult to be collected and could also move into deeper soil layer, posed difficulties for MaPs recycle. In addition, combined with the strong winds in Xinjiang (*Xiong et al., 2019*), highly fragmented plastic debris in fields could be easily transferred to other environments by the wind (*He et al., 2018*; *Rezaei et al., 2019*; *Steinmetz et al., 2016*). The residual ratios measured in our two study regions suggested that the longer the plastic debris remained in the fields, the more likely that the plastic would disperse to other environments, which

would affect the plastic debris accumulation and pose a threat to the environment. Overall, the accumulation pattern of MaPs in Xinjiang, a high machinery intensity region, subjected to many factors. Thus, the nonlinear increase of MaPs raises an important question: do other natural factors have significant effects on agricultural plastic accumulation? If so, what is the relative importance of these different factors?

According to our results, in both two study regions, in general, MaPs number in 0–10 cm soil showed no or less significant difference compared to 10–20 cm, however, as for MaPs mass, 0–10 cm and 10–20 cm layers usually showed significant difference. This result indicated that even though 10–20 cm soil contained less amount MaPs compared to 0–10 cm soil, it still contained a significant MaPs number. Previous studies in China (*Ma et al., 2008*; *Yan et al., 2008*) have indicated that long-term tillage and intense machine tillage/ploughing might have homogenized the soil, especially in the top 0–20 cm, thus leading to the insignificant differences seen for MaPs number among the various layers. This result also suggested that MaPs number (p·ha$^{-1}$) should also be an indicator for plastic pollution in future research. Machinery tillage intensity can also affect the size of the MaPs in soils. In our research, the majority size categories of the MaPs collected in S1 were 10–50 cm$^2$ and 2–10 cm$^2$, while for S2, the majority of collected MaPs were 0.25–2 cm$^2$ and 2–10 cm$^2$. These results indicate that the MaPs in S2 were more fragmented as compared to S1. These results agree with previous research findings that the sizes of plastic debris found in regions where applied with low-intensity machinery tillage are usually bigger than in regions where applied with high-intensity machinery tillage (*Li et al., 2017*; *Ma et al., 2008*; *Yan et al., 2008*).

## Microplastics in agricultural soils

In current research, MiPs were mainly detected in the soils exposed to 30 years of mulching history in S1 and were detected in all the selected fields in S2. These results are in agree with the MaPs number percentage results that in S2, MaPs were more fragmented than in S1. These results also indicated that long-term exposure of plastic debris in agricultural fields and high-intensity machine tillage could create more ubiquitous MiPs. Our MiPs results were far more less than reported in other studies. Research conducted in southwestern China reported that MiPs were detected in the range of 71–429 p·10g$^{-1}$ in the 0–10 cm layer of soil in a vegetable production system housed in a plastic greenhouse (*Zhang & Liu, 2018*). They attributed the higher MiPs to the intense use of wastewater and sewage irrigation needed for the intensive vegetable rotation (6–8 crops per year). However, in our research, the cropping rotation and irrigation intensity were less than those vegetable fields.

The rare MiPs detected in current research might be attributed to the extraction method limitation mentioned in Material and method. However, the MiPs data in our research could be regarded as a minimum estimation of accumulation of LDPE sourced MiPs in agricultural fields, our work has made the attempt to connect MiPs pollution to plastic mulching use in a real in situ study. More detailed research with better detection methods need to take place for a good estimation of the amount of MiPs in the soil profile.

## CONCLUSIONS

In this paper, we have shown that different farming systems can affected accumulation and distribution of agricultural plastic debris (both MaPs and MiPs). Our study confirmed our hypothesis that (1) under the same farming system (low-intensity machinery tillage), continuous mulching could accumulate more MaPs than intermittent mulching; (2) high-intensity machinery tillage farming system (S2) could lead to higher fragmentation of MaPs and lead to higher fragmentation of MaPs and a create severer MiPs pollution as compared to low-intensity machine tillage farming systems (S1). We also found that in S1, crop rotation system could affect ploughing time (Spring or Autumn), thus affecting the accumulation of MaPs. The residual ratios were lower for fields with a long mulching history. However, it remains unclear if this is due to wind and/or water transportation or due to further degradation of MaPs into smaller particles or even MiPs, which are difficult to recycle. Further research on the degradation process of agricultural plastic debris are needed, which could also provide a better understanding of the risk of agricultural MaPs and MiPs and its effects on soil health and food quality.

## ACKNOWLEDGEMENTS

Many thanks to Yi Dang from Gansu Academy of Agricultural Sciences and Yu Liu from Xinjiang Academy of Agricultural Sciences for helping to facilitate contact with local farmers and for providing detailed information of local farming practices and plastic film specification. Thanks to Demie Moore for checking and editing the language of this manuscript. Thanks to Nicolas Beriot for providing the picture of setup of microplastic identification.

### Funding

Funding for this research came from the Key Laboratory of Efficiency of Water Utilization in Dryland Farming, Dryland Agriculture Institute, Gansu Academy of Agricultural Sciences, China (HNSJJ-2019-03, HNSJJ-2019-04), the EU Horizon 2020 project (ISQAPER: 635750) and the National Key Research and Development Program of China (Grant No. 2016YFE0112700) and the China Scholarship Council (CSC: 201707720007). The funders had no role in study design, data collection and analysis, decision to publish, or preparation of the manuscript.

### Grant Disclosures

The following grant information was disclosed by the authors:
Key Laboratory of Efficiency of Water Utilization in Dryland Farming, Dryland Agriculture Institute, Gansu Academy of Agricultural Sciences, China: HNSJJ-2019-03 and HNSJJ-2019-04.
EU Horizon 2020 project (ISQAPER): 635750.

National Key Research and Development Program of China: 2016YFE0112700.
China Scholarship Council (CSC): 201707720007.

## Competing Interests

The authors declare that they have no competing interests. Violette Geissen is an Academic Editor for PeerJ.

## Author Contributions

- Fanrong Meng conceived and designed the experiments, performed the experiments, analyzed the data, prepared figures and/or tables, and approved the final draft.
- Tinglu Fan conceived and designed the experiments, authored or reviewed drafts of the paper, and approved the final draft.
- Xiaomei Yang conceived and designed the experiments, authored or reviewed drafts of the paper, and approved the final draft.
- Michel Riksen conceived and designed the experiments, authored or reviewed drafts of the paper, and approved the final draft.
- Minggang Xu conceived and designed the experiments, authored or reviewed drafts of the paper, and approved the final draft.
- Violette Geissen conceived and designed the experiments, authored or reviewed drafts of the paper, and approved the final draft.

## Field Study Permissions

The following information was supplied relating to field study approvals (i.e., approving body and any reference numbers):

Co-author Prof.Tinglu Fan had good contact with local farmers, the following farmers provided permission to allow us to conduct the field observation: Yi Dang, Shangzhong Li, Lei Wang, Limin Wang, Sanzhi LI, Yu Liu, Gongmao Wang, Jihong Shi and Jiancheng Liu.

## Data Availability

The raw data are available in the Supplemental Files.

## Supplemental Information

Supplemental information for this article can be found online at http://dx.doi.org/10.7717/peerj.10375#supplemental-information.

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
