# Peer review of "Effects of plastic mulching on the accumulation and distribution of macro and micro plastics in soils of two farming systems in Northwest China"

_PeerJ, doi:10.7717/peerj.10375_

## Round 0.1 · original submission · Major Revisions

As commented by the reviewers. The manuscript addresses an interesting topic but requires major revisions. Kindly attend to all comments raised by the reviewers before resubmitting the manuscript.

Reviewer 1 ·

Basic reporting

no comment

Experimental design

The method of MiPs separation is not correct at all. The flotation separation method used in this paper comes from "zhang, S., Yang, X., Gertsen, H., Peters, P., Salanki, T., Geissen, V., 2018. A simple method for the extraction and identification of light density microplastics from soil. Sci Total Environ 616-617, 1056-1065". In fact, the method given in this paper is not to separate plastic film, but plastic particles or fibres. The tiny plastic film in the soil is always wrapped with soil particles, and it is impossible for distilled water to float them out,and at least, saturated salt solution must be used for flotation agent. It is possible that the use of incorrect flotation solution resulted in large deviation in MIPS data and no MiPs found in most of those samples. So, based on the uncertainty of experimental method, MiPs data is not credible.

Validity of the findings

this is a great topic

Additional comments

The manuscript "Effect plastic mulching on the accumulation and distribution of macro and micro plastics in soil - A case study of two farming systems in Northwest China" investigated the accumulation of plastics in the agricultural soils. For now, most of the studies have estimated the occurrence of microplastics in aquatic environments. Very few studies have talked about the terrestrial environments. From this point, this manuscript has some novelty. However, the modification of the writing English is highly needed. The logic of the results was not clear. Why the accumulation and distribution were shown separately? It will be better to organize one section for macroplastics, and one section for microplastics. Additionally, this no information about the amount of microplastics in abstract. Also, there is no need to tell the categorization of macroplastics in the abstract. Overall, this is a great topic, and the authors should organize this manuscript better before it can be accepted.

There was no line numbers in this manuscript.

Title
There is no need to emphasize that this work is a case study.

Abstract—it will be better to re-write the abstract
The grammar of the first sentence was quite strange, please modify this sentence.

From the first 0-30 cm? from the top

Introduction
Modify the first sentence.

Modify "Plastic mulch proves" to previous studies proved or it has been proved that

Using plastic mulch ----plastic mulching

There is a growing concern

Not only these studies have reported the accumulation of plastic debris in the farming lands. For instance, Huang et al. 2020, Environmental Pollution "Agricultural plastic mulching as a source of microplastics in the terrestrial environment"

There is no clear hypothesis in the introduction. That may be the potential reason that this manuscript looks confusion. The author should talk about the farming practices such as the level of mechanization, mulching pattern, significantly influence the profiles of plastic residues in the soil.

It is very impressive that the authors showed the details of study area.

For the identification of microplastics, any references? If the authors developed this method, the limit of detection and the recovery should be added.

The author said it was the wind dispersed agricultural plastic debris to other environments. This reason seems not very robust. More discussion is needed.

It will be better if the author can give the FTIR results.

·

Basic reporting

The manuscript is well written and suitable for their publication. It covers an interesting topic about macro- and microplastics in agricultural soils, one of the primary sources of plastics in terrestrial ecosystems. However, in my opinion, the manuscript needs several improvements (main attention for question 12, 12.1 and 13).

1. I like the supplementary data file. It's very good. However, in my opinion, Table 1 should be as Table S1 (Supplementary data), and Tables S2, 4, 6 and 8 should be moved to the manuscript and not as Supplementary data. Besides, I saw that Table S2 and Table S6 is the same information that is written in Figures 3 and 5. Is it right? I suggest to avoid duplicate information. In my opinion, it should be more useful to left the tables for comparison studies, and delete the figures. Besides, Figure S1 should be added to Figure 2 (merged in the same figure).

Experimental design

2. L96-101 I don't understand the differentiation between <2 and > 0.25 cm2, because Microplastics are also called to particles less than 5 mm, and since some years ago, MiPs are called to particles less than 1 mm and not 2 mm. I understand that < 2 mm was chosen due to particle soil size, but in my opinion, these questions should be clarified and improved. Also, a density of less than 1 g cm3 was chosen by authors, why? Which criteria was followed?

3. L105. Which farming system? Greenhouses? Open cultures? Which cultivation (Strawberry?)? These sentences (and also 109-110 lines) should be improved, or even give some pictures in the manuscript or Supplementary data. This information is given in lines 121... and for this reason, I suggest to improve this section. For example, lines 121- should be moved to 108-. It should be more readable, give first general information and after give information for each soil.

4. L124-127. I don't understand why plastic mulch is used for maize crop. Is Early sweet corn that needs plastic mulch to improve their grown? https://doi.org/10.21273/HORTSCI14667-19 https://lib.dr.iastate.edu/farms_reports/521/ Besides, how is removed the plastic mulch? Mechanically? Handmade? By wind?

5. L150. "MaPs were manually collected using a quadrat sampling method. For each sampling field, we randomly selected three quadrats 100 cm long, 50 cm wide and 30 cm deep which covered two crop rows (Fig. S1)" Which area was sampled? 1 Ha? How many samples by each study area? Improve this section. I suggest add a figure for the manuscript or to the Supplementary Information.

6. L153. I think that bag showed in Fig S2 is a PP mesh bag with probable plastic contamination to samples. Besides, why the samples were not sieved directly in the field?

7. L162-163. This method is new, or authors are following a published methodology? I suggest add also a citation as indicated by Line 199

8. L195. "polypropylene (PP) plastic bags." Figure S2b, 2c?

9. "plastic-free room" How? Please, explain this.

10. "A control field where no plastic mulch had ever been used was selected from each study site." Microplastics can be carried by wind, car tyres (agricultural machinery as a source of soil contamination), or even by fertilizers. I suggest improving this explanation.

11. Keywords: I suggest to add the plastic polymer (low-density polyethene) and not macro- and micro-plastics because they are in the title.

Validity of the findings

12. I like the proposed method for microplastics extraction due to it's simple and very easy to use for non-resources laboratories. However, some authors also proposed to add Fenton Reagent or KOH for organic matter digestion. Considering that these are agricultural soils, and the organic matter content must be higher than 2%, why was no further digestion of organic matter added? See https://doi.org/10.1021/acs.est.9b04618 or https://doi.org/10.1021/acs.est.8b01517 Unnecessary? I suggest to improve this information. Besides; some authors also make a plastic identification by FTIR. Why was not used in this plastic identification and only by optical microscopy? Unnecessary?

12.1. I understand (lines 387-401) that authors do not have the facilities for organic matter digestion or high-density plastic extraction (only plastic with density less than 1 g cm3 were extracted). I suggest to add this information in the material and methods to show the limitations of the present study because this is a significant limitation of the present study. No more source of plastics in these sources.

I suggest to improve this question in the material and methods, or even in the introduction (e.g. line 100), to justify why high-density plastics were not extracted.

13. I did not see in the results and discussion information about the shapes (or colour) of Macro and microplastics. All plastic items were found as fibres or undefined shapes? If possible, please add this information.

---

## Round 0.2 · accepted · Accept

All comments raised by the reviewers were answered and the manuscript in their opinion has improved tremendously.

Reviewer 1 ·

Basic reporting

The authors have responsed all the comments.

Experimental design

The authors have responsed all the comments.

Validity of the findings

The authors have responsed all the comments.

Additional comments

This manuscript can be published now.

·

Basic reporting

The submitted paper has been improved since last submission. Now, several sections are more clear and easy to read. However, i have my doubts regarding to extraction method, although this question was clarified and improved.

I think that all questions were addressed.

Experimental design

No comment.

Validity of the findings

I have my doubts regarding to extraction method, although this question was clarified and improved. However, I think that all questions were addressed. The paper is interesting.

Additional comments

The submitted paper has been improved since last submission. Now, several sections are more clear and easy to read. However, i have my doubts regarding to extraction method, although this question was clarified and improved.

I think that all questions were addressed.